# Negative Survival Impact of Occult Lymph Node Involvement in Small HER2-Positive Early Breast Cancer Treated by Up-Front Surgery

**DOI:** 10.3390/cancers15184567

**Published:** 2023-09-14

**Authors:** Gilles Houvenaeghel, Monique Cohen, Marc Martino, Fabien Reyal, Jean-Marc Classe, Marie-Pierre Chauvet, Pierre-Emmanuel Colombo, Mellie Heinemann, Eva Jouve, Pierre Gimbergues, Anne-Sophie Azuar, Charles Coutant, Anthony Gonçalves, Alexandre de Nonneville

**Affiliations:** 1Department of Surgical Oncology, CRCM, Institut Paoli-Calmettes, Aix-Marseille University, CNRS, INSERM, 232 Bd de Sainte Marguerite, 13009 Marseille, France; cohenm@ipc.unicancer.fr (M.C.); martinom@ipc.unicancer.fr (M.M.); 2Institut Curie, 26 Rue d’Ulm, 75248 Paris, France; fabien.reyal@curie.fr; 3Institut René Gauducheau, Site Hospitalier Nord, Boulevard Jacques Monod, 44800 St. Herblain, France; jean-marc.classe@ico.unicancer.fr; 4Centre Oscar Lambret, 3 Rue Frédéric Combenal, 59000 Lille, France; mp-chauvet@o-lambret.fr; 5Centre Val d’Aurelle, 31 Rue de la Croix Verte, 34090 Montpellier, France; pierre-emmanuel.colombo@icm.unicancer.fr; 6Centre Léon Bérard, 28 Rue Laennec, 69008 Lyon, France; mellie.heinemann@lyon.unicancer.fr; 7Centre Claudius Regaud, 20-24 Rue du Pont St. Pierre, 31300 Toulouse, France; jouve.eva@iuct-oncopole.fr; 8Centre Jean Perrin, 58 Rue Montalembert, 63011 Clermont-Ferrand, France; pierre.gimbergues@cjp.fr; 9Hôpital de Grasse, Chemin de Clavary, 06130 Grasse, France; as-azuar@chgrasse.fr; 10Centre Georges François Leclerc, 1 Rue du Professeur Marion, 21000 Dijon, France; ccoutant@cgfl.fr; 11Department of Medical Oncology, CRCM, Institut Paoli-Calmettes, Aix-Marseille University, CNRS, INSERM, 232 Bd de Sainte Marguerite, 13009 Marseille, France; goncalvesa@ipc.unicancer.fr

**Keywords:** breast cancer, HER2-positive, sentinel node, micro-metastases, survival

## Abstract

**Simple Summary:**

Our objective was to investigate the impact of pN0(i+) or pN1mi in HER2-positive breast cancer patients undergoing up-front surgery on their outcomes. Survival was not adversely affected by pN0(i+) and pN1mi in 1771 HER2-positive patients. However, in the case of pT1a-b HER2-positive breast cancers, a negative impact on recurrence-free survival was observed specifically for patients with pN0(i+) and pN1mi diseases, particularly among those with pT1b tumors without adjuvant chemotherapy. Our findings highlight the importance of considering the pN0(i+) and pN1mi status in the decision-making process when discussing trastuzumab-based adjuvant chemotherapy for these patients.

**Abstract:**

(1) Background: The independent negative prognostic value of isolated tumor cells or micro-metastases in axillary lymph nodes has been established in triple-negative breast cancers (BC). However, the prognostic significance of pN0(i+) or pN1mi in HER2-positive BCs treated by primary surgery remains unexplored. Therefore, our objective was to investigate the impact of pN0(i+) or pN1mi in HER2-positive BC patients undergoing up-front surgery on their outcomes. (2) Methods: We retrospectively analyzed 23,650 patients treated in 13 French cancer centers from 1991 to 2013. pN status was categorized as pN0, pN0(i+), pN1mi, and pNmacro. The effect of pN0(i+) or pN1mi on outcomes was investigated both in the entire cohort of patients and in pT1a-b tumors. (3) Results: Of 1771 HER2-positive BC patients included, pN status distributed as follows: 1047 pN0 (59.1%), 60 pN0(i+) (3.4%), 118 pN1mi (6.7%), and 546 pN1 macro-metastases (30.8%). pN status was significantly associated with sentinel lymph node biopsy, axillary lymph node dissection, age, ER status, tumor grade, and size, lymphovascular invasion, adjuvant systemic therapy (ACt), and radiation therapy. With 61 months median follow-up (mean 63.2; CI 95% 61.5–64.9), only pN1 with macro-metastases was independently associated with a negative impact on overall, disease-free, recurrence-free, and metastasis-free survivals in multivariate analysis. In the pT1a-b subgroup including 474 patients, RFS was significantly decreased in multivariate analysis for pT1b BC without ACt (HR 2.365, 1.04–5.36, *p* = 0.039) and for pN0(i+)/pN1mi patients (HR 2.518, 1.03–6.14, *p* = 0.042). (4) Conclusions: Survival outcomes were not adversely affected by pN0(i+) and pN1mi in patients with HER2-positive BC. However, in the case of pT1a-b HER2-positive BC, a negative impact on RFS was observed specifically for patients with pN0(i+) and pN1mi diseases, particularly among those with pT1b tumors without ACt. Our findings highlight the importance of considering the pN0(i+) and pN1mi status in the decision-making process when discussing trastuzumab-based ACt for these patients.

## 1. Introduction

The prognostic value of axillary lymph node invasion by isolated tumor cells (ITC) or micro-metastases has been the subject of numerous studies with divergent results but without any distinction between breast cancer (BC) molecular subtypes [1]. Since the introduction of sentinel lymph node biopsy (SLNB), ITC and micro-metastases have been detected more often in patients with BC. This limited metastatic lymph node involvement is observed in 8–10% of patients with early BC and sentinel lymph node biopsy (SLNB), representing 10–28% of patients with involved sentinel node [1,2,3,4,5,6,7,8,9,10,11,12,13,14,15,16,17,18,19,20,21,22,23,24,25,26,27,28,29]. Immuno-histo chemistry (IHC) analysis increased the SN involvement rate from 9% to 47% when compared with HES only [30]. However, different rates of LN involvement according to tumor subtypes were reported with lower rates in triple-negative BC and higher rates in HER2-positive BC [31,32,33]. The presence of ITC or micro-metastases in the axillary lymph nodes of triple-negative cancers has an independent negative prognostic value, particularly in association with the presence of LVI in patients treated by up-front surgery [34,35]. Conversely, no independent negative prognostic value has been shown for the presence of ITC or micro-metastases in the axillary lymph nodes of endocrine receptor (ER)-positive HER2-negative cancers [36]. The prognostic value of ITC or micro-metastatic axillary lymph node involvement in HER2-positive BCs treated by primary surgery has not been specifically studied. HER2-positive BCs larger than 2 cm or with clinical and/or ultrasound involvement of the axillary lymph nodes are currently treated with neoadjuvant chemotherapy [37], except where contraindicated by age, comorbidities, and, in particular, physiological age. However, as the rate of complete pathological response does not correlate with the initial clinical size of the tumor, neoadjuvant chemotherapy is increasingly proposed for tumors with no clinical axillary lymph node involvement of more than 15 mm, or even more than 10 mm [38]. This study aimed to determine, from a multicenter cohort, the prognostic value of axillary node invasion by ITC or micro-metastases in HER2-positive BCs treated by primary surgery for all patients, as well as for patients with pT1a-b cancer.

## 2. Methods

### 2.1. Study Design and Data Source

The medical records of 23,650 patients that were treated from January 1991 to December 2013 were retrieved from the clinical databases of 13 cancer centers in France for retrospective analysis. Of this initial cohort, all patients treated with primary surgery for HER2-positive BC, with or without adjuvant chemotherapy and trastuzumab, who had undergone breast conservative surgery (BCS) or mastectomy, were included. Data were collected on patient and tumor characteristics, treatments received, and clinical outcomes.

### 2.2. Pathological Assessment

The determination of ER and HER2 status followed national guidelines, where estrogen and/or progesterone receptor positivity was assessed using IHC with a 10% threshold for ER positivity. HER2 positivity was identified by either a 3+ IHC score or HER2 amplification detected through in situ hybridization. To determine lymphovascular invasion (LVI), trained pathologists examined HES slides and identified the presence of lymphovascular emboli, characterized as tumor cells within an endothelium-lined space in the peritumoral area [39]. All sentinel lymph node biopsies were analyzed by serial sections with standard HES after fixation. No intraoperatively analysis were performed. If all the serial sections were negative, an additional IHC analysis was carried out.

### 2.3. Statistical Analysis

Analyses were performed separately for all patients and by ER status, on factors associated with pN status (categorized in four groups as pN0, pN0(i+), pN1mi, and pNmacro—defined as any pN+ greater than 2 mm) according to patient, disease, and clinical characteristics such as age, tumor size, Scarff–Bloom–Richardson (SBR) grade, IHC surrogate of molecular subtypes luminal B-like/HER2-positive and Her-positive/ER-negative, breast and axillary surgery, endocrine therapy (ET), adjuvant chemotherapy (ACt), and radiotherapy. Overall survival (OS) was defined as the time interval from the date of surgery to death or last follow-up; disease-free survival (DFS) was defined as the time interval from the date of surgery to any event (recurrence, metastasis, or death) or last follow-up; recurrence-free survival (RFS) was defined as the time interval from the date of surgery to local, regional, or distant recurrence whichever comes first or last follow-up; metastasis-free survival (MFS) was defined as the time interval from the date of surgery to distant recurrence or death as a first event or last follow-up. Patients lost to follow-up were considered as alive as of the date of last contact. The associations between categorical values were evaluated via χ^2^ tests. Factors significantly associated with pN status were determined by binary logistic regression adjusted for all significant variables determined by univariate analysis. Kaplan–Meier method and log-rank tests were employed to analyze survival functions, and multivariate survival analyses were performed using the Cox proportional-hazard regression model adjusted for significant variables. Subsequent analyses focused on patients with pT1a-b pN0-pN0(i+) or pN1mi. A significance level of *p* ≤ 0.05 was set, and SPSS 16.0 (SPSS Inc., Chicago, IL, USA) was used for all analyses. All procedures performed in this study involving human participants were carried out by French ethical standards and with the 2008 Helsinki Declaration. As this was a retrospective non-interventional study, no formal personal consent was required. Authorization to use the database was obtained from the strategic orientation committee of the Paoli-Calmettes Institute (ClinicalTrials.gov NCT02869607).

## 3. Results

### 3.1. Association of pN Status with Other Clinical and Pathological Features

HER2-positivity for patients treated by up-front surgery was reported in 1771 cases: 1047 pN0 (59.1%), 60 pN0(i+) (3.4%), 118 pN1mi (6.7%), and 546 pN1 with macro-metastases (30.8%). Pathologic tumor size was less than 20 mm for 1082 patients (61.1%). Median age was 55 years (mean 55.64; CI 95% 55.1–56.2). Median tumor size was 16.0 mm (mean 20.1; CI 95% 19.3–21.0): 13.0 (mean 15.1), 15.0 (mean 17.1), 16.0 (mean 18.9), and 23.5 (mean 29.9) for pN0, pN0(i+), pN1mi, and pN1 with macro-metastases, respectively. pN status was significantly associated with clinical and pathological characteristics (Table 1): sentinel lymph node biopsy (SLNB), axillary lymph node dissection (ALND), age groups, endocrine receptors status, SBR grade, tumor size < or ≥20 mm, lymphovascular invasion (LVI), type of surgery, adjuvant chemotherapy with trastuzumab, radiotherapy or not, regional node irradiation (RNI) (Table 1).

In logistic regression analysis, pN status was significantly associated with pT size ≥ 20 mm (*p* < 0.0001), ER status (*p* = 0.005), and LVI (*p* < 0.0001). Age groups and SBR grade were not statistically associated with pN status.

For patients with known LVI status, LVI was present in 33.2% (530/1594) of cases: 18.6% (170/915) of pN0, 43.8% (25/57) of pN0(i+), 46.0% (52/113) of pN1mi, and 55.6% (283/509) of pN1 macro-metastases. Similar results were observed by ER status: 17.1% (48/280), 47.4% (9/19), 46.9% (15/32), 56.8% (117/206) (*p* < 0.0001), and 19.2% (122/634), 42.1% (16/38), 45.7% (37/81), 54.8% (166/303) (*p* < 0.0001), in ER-negative and ER-positive BC respectively.

ACt was administered in 67.0% of pN0 patients (702/1047), 81.7% of pN0(i+) (49/60), 93.2% of pN1mi (110/118), and 93.8% of pN1 with macro-metastases (512/546) (*p* < 0.0001). In binary logistic regression analysis, ACt was significantly associated with pT ≥ 20 mm (odds ratio, OR 2.476, CI 95% 1.74–3.52, *p* < 0.0001), grade 2 (OR 3.288, 2.15–5.03, *p* < 0.0001), grade 3 (OR 10.040, 6.26–16.10, *p* < 0.0001), LVI (OR 2.066, 1.39–3.06, *p* < 0.0001), ER-positivity (OR 0.739, 0.55–1.00, *p* = 0.050), age >40–50 (OR 0.498, 0.28–0.89, *p* = 0.019), age >50–74.9 (OR 0.327, 0.19–0.56, *p* < 0.0001), and age ≥ 75 (OR 0.044, 0.02–0.09, *p* < 0.0001) compared to an age ≤40, as well as pN status: pN1mi (OR 6.727, 3.03–14.94, *p* < 0.0001) and pN1 with macro-metastases (OR 4.657, 3.04–7.13, *p* < 0.0001) compared to pN0 status; pN0(i+) was not statistically associated with ACt (OR 1.582, 0.76–3.28, *p* = 0.219) (Appendix A).

### 3.2. Prognostic Impact of pN Status on OS, DFS, RFS, MFS in the Entire Population: Univariate Analysis

Median follow-up was 61 months (mean 63.2; CI 95% 61.5–64.9): 58.8 (mean 59.9), 62.7 (mean 61.7), 67 (mean 68.9), and 64.0 (mean 68.3) for pN0, pN0(i+), pN1mi, and pN1 with macro-metastases, respectively.

OS was significantly associated with pN status (*p* < 0.0001): the 2, 5, 7, and 10-year OS were 98.9%, 95.5%, 93.0%, and 88.4% for pN0, 96.6% at 2-5-7 years for pN0(i+), 100%, 98.0%, 95.7%, and 95.7% for pN1mi, and 97.3%, 88.2%, 82.9%, and 76.8% for pN1 with macro-metastases. Several criteria were associated with lower OS: ER-negativity (*p* = 0.008), mastectomy (*p* < 0.0001), age (*p* < 0.0001), pT ≥ 20 mm (*p* < 0.0001), and LVI (*p* < 0.0001). Grade and ACt were not statistically associated with OS (*p* = 0.636 and 0.371, respectively).

### 3.3. Prognostic Impact of pN Status on OS, DFS, RFS, MFS in the Entire Population: Multivariate Analyses

Only pN1 with macro-metastases was significantly associated with a negative impact on OS (HR 1.583, 1.01–2.49, *p* = 0.048), DFS (HR 1.737, 1.25–2.41, *p* = 0.001), RFS (HR 2.044, 1.42–2.95, *p* < 0.0001), and MFS (HR 1.824, 1.26–2.64, *p* = 0.001). Other significant factors associated with worse OS and DFS were LVI, pT ≥ 20 mm, ER-negativity, absence of ACt, and age ≥ 75 years (Table 2).

### 3.4. Subgroup Analysis for pT1a-b Tumors with pN0 or (pN0(i+) and pN1mi)

This subgroup analysis included 474 patients, 178 pT1a (37.6%), and 296 pT1b (62.4%) with 426 pN0 (89.9%), 20 pN0(i+) (4.2%), and 28 pN1mi (5.9%). Characteristics of patients according to pN0 and pN0(i+) or pN1mi are reported in Table 3. A statistically significant difference was observed for ACt, type of surgery, age groups, and LVI. SBR grade and ER status were not statistically different.

ACt was statistically significantly associated with SBR grade, ER status, age groups, LVI, and pN status (Appendix A). The type of surgery did not impact ACt use. In a binary logistic regression analysis, ACt was statistically significantly associated with SBR grade, ER status, age groups, LVI, and pT1. Statistical difference for pN status was not reached but a trend was observed (Table 4).

At a median follow-up of 59.8 months (mean 62.7), the following events had occurred: 22 deaths, 39 recurrences, 52 death or recurrences, and 16 metastases. In univariate analysis, pN status, pT size, type of surgery, ER status, and grade were non-statistically significantly associated with OS. LVI, age, and no ACt were significantly associated with decreased OS. In multivariate analysis adjusted on ACt, ER-status, age groups, LVI, pT size, and pN status (patients with pN0(i+) or pN1mi) were associated with decreased RFS (HR 2.548, CI 95% 1.06–6.13, *p* = 0.037) and patients with ACt had better RFS (HR 0.288, 0.13–0.65, *p* = 0.003) (Figure 1, Table 5). pN status was not associated with a difference in OS and DFS.

ACt was associated with pT and LVI status: 36% for pT1a/no LVI (45/125), 55.6% for pT1a/LVI+ (5/9), 61.2% for pT1b/no LVI (131/214), and 92.7% for pT1b/LVI+ (38/41) (*p* < 0.0001). In multivariate analysis, RFS was decreased in pT1b tumors without LVI (HR 2.893, CI 95% 1.09–7.67, *p* = 0.033), in patients without ACt (HR 3.758, 1.41–9.99, *p* = 0.008), in patients with ER-negative BC (HR 2.702, 1.10–6.63, *p* = 0.030), and patients with pN0(i+) and pN1mi (HR 2.780, 1.10–7.04, *p* = 0.031) (Figure 2). A better RFS was observed for patients over 50 and under 75 years (HR 0.189, 0.06–0.55, *p* = 0.002).

In multivariate analysis adjusted on LVI, ER status, age groups, pN status, and pT size with or without ACt, RFS was significantly decreased for pT1b BC without ACt (HR 2.365, 1.04–5.36, *p* = 0.039) and for pN0(i+) or pN1mi (HR 2.518, 1.03–6.14, *p* = 0.042) (Figure 3).

For patients with pT1a BC, no significant difference was observed between patients stratified according to ACt and pN status. Patients with pN0(i+) or pN1mi had a lower RFS but without a statistical difference (only nine patients in this later group) (Figure 4).

For patients with pT1b BC, only patients with pN0 and ACt exhibit better RFS (Figure 5): HR 0.140, 0.04–0.48, *p* = 0.002 in comparison with patients with pN0 without ACt.

## 4. Discussion

The main result of this study is the absence of prognostic value of pN0(i+) or pN1mi in HER2-positive BC in the whole cohort, but a negative impact on RFS in pT1a-b patients, particularly for those with pT1b tumors. Our results support the consideration of pN0(i+) and pN1mi status in the decision-making process when discussing trastuzumab-based ACt for these patients.

### 4.1. Incidence of pN0(i+) and pN1mi and Non-Sentinel Node (NSN) Involvement Rates

In a study reported by Reyal et al. [33], positive SN rates were 34.7% (439/1264) for ER-positive/HER2-negative tumors, 31.6% (25/79) for ER-positive/HER2-positive tumors, 41.5% (22/53) for ER-negative/HER2-positive tumors, and 20.4% (30/147) for triple-negative tumors, but without distinction between pN0(i+), pN1mi, and pN1-macro-metastases (Appendix A). In the French cohort of 12,572 patients with SLNB [40], pN1mi rate was 8% (*n* = 970) and pN0(i+) was 3% (*n* = 355) with 66% of pN0 (*n* = 8253) and 24% of pN-macro-metastases (*n* = 2994). By tumor subtypes, pN0(i+) and pN1mi rates were 11.08% (996/8988) and 34.72% (996/2869) among all patients and among patients with involved lymph nodes and Luminal-A-like tumors, 12.64% (149/1178) and 24.96% (149/597) among patients with Luminal-B-HER2-negative-like tumors, 8.88% (68/766) and 23.45% (68/290) among patients with ER-positive/HER2-positive tumors, 8.89% (40/450) and 19.42% (40/206) among patients with ER-negative/HER2-positive tumors, and 6.42% (70/1091) and 20.71% (70/338) among patients with triple-negative tumors. In the SERC trial [41] that includes patients with SN involvement, SN ITC was present in 5.91% of patients, micro-metastases in 28.12%, and macro-metastases in 65.97%. According to tumor phenotypes, ITC, micro-metastases, and macro-metastases rates were respectively 5.5% (75/1354), 27.5% (373/1354), and 66.9% (906/1354) among patients with ER-positive/HER2-negative tumors, 6.67% (9/135), 27.4% (37/135), and 65.9% (89/135) among patients with ER-positive/HER2-positive tumors, 7.3% (3/41), 24.4% (10/41), and 68.3% (28/41) among patients with ER-negative/HER2-positive tumors, and 8.2% (8/97), 29.9% (29/97), and 61.8% (60/97) among patients with triple-negative tumors. Crude rates of positive NSN according to SN status were 6.1% for patients with ITC (2/33), 10.3% for SN micro-metastases (22/214), and 25.7% for SN macro-metastases (134/522). Positive-NSN rates in multivariate analysis for patients with completion ALND were statistically significantly associated with ACt when performed after ALND (OR 2.99, *p* < 0.0001) in comparison to the absence of ACt and non-statistically significant when ACt was performed before ALND (OR 1.51, *p* = 0.232), in comparison with the absence of ACt use. No differences between ITC, micro-metastases, and macro-metastases SN status were reported.

In summary, there was no strong difference in ITC and micro-metastases rates and no strong difference in NSN rates, according to tumor subtypes. However, a strong downstaging on NSN involvement rate was reported when completion ALND was performed after ACt, whatever tumor subtype.

### 4.2. Prognostic Value of Micro-Metastases in Triple-Negative and ER-Positive/HER2-Negative Patients

Axillary lymph node involvement is a key prognostic feature in early TNBC when ITC and micro-metastases are identified. The independent prognostic features identified for DFS were tumor size ≥ 20 mm (HR = 1.86; 95% CI: 1.11–3.10, *p* = 0.018), LVI (HR = 1.69; 95% CI: 1.21–2.34, *p* = 0.002), and axillary lymph node involvement both in case of macro-metastases (HR = 1.97; 95% CI: 1.38–2.81, *p* < 0.0001) and occult metastases (HR = 1.72; 95% CI: 1.1–2.71, *p* = 0.019) [34]. In a recent study, [36] including 13,773 ER-positive/HER2-negative patients, with 546 pN0(i+) and 1446 pN1mi cases, LN micro-metastases had no detectable prognostic impact leading to the conclusion that they should not be considered as a determining factor in indicating ACt. This observation was also suggested in other recent studies [21,23,24].

#### Indication of ACt and Trastuzumab in HER2-Positive BC

Axillary micro-metastases do not impact the indication of adjuvant treatment with chemotherapy ± trastuzumab in HER2-positive and triple-negative tumors with pathologic size >2 cm, and even > 10 mm [42,43]. Neo-adjuvant chemotherapy is usually administered for patients with pT2 or pT1a-b-c N1 BC [37]. ACt after up-front surgery is indicated for patients with ≥ pT1c or pN1 macro-metastases but should also be discussed for patients with pT1a-b pN0 or with ITC or micro-metastases.

### 4.3. Available Literature on HER2-Positive, pT1ab BC

Adjuvant trastuzumab-based chemotherapy clearly improved survival in randomized clinical trials [44,45,46,47], but there were few or no patients with pT1abN0 stages and this specific situation was only addressed in relatively rare retrospective cohort studies [48,49,50,51,52,53]. In our study [42], ACt ± trastuzumab was associated with a significantly reduced risk of recurrence, including distant recurrence, in infra-centimetric node-negative HER2-positive BC and most of the benefit may have been driven by pT1b tumors.

No data were identified specifically for pT1a-b, HER2-positive, pN0(i+), or pN1mi in our literature search.

### 4.4. Negative Impact of LVI

A strong interaction between the presence of LVI and axillary lymph node involvement was reported whatever tumor subtypes [40] and both for ER-positive and ER-negative tumors [39]. In the present study, for patients with HER2-positive tumors and known LVI status, LVI was present in 33.2% (530/1594): 18.6% (170/915) of pN0, 43.8% (25/57) of pN0(i+), 46.0% (52/113) of pN1mi, and 55.6% (283/509) of pN1 macro-metastases.

In our recent study [39], LVI was present in 24% (4205/17,322) of patients and was most prevalent in patients with luminal B-like tumors, specifically high-grade HER2-negative tumors (44%)—22% (3279/14,655) of ER-positive and 35% (926/2667) of ER-negative tumors—(Appendix A). In the present study, LVI was present in 33.2% (530/1594) of HER2-positive BC—35.2% (189/537) of ER-positive and 32.3% (341/1056) of ER-negative tumors In the subgroup of pT1a-b BC, LVI was present in 10.5% (36/344) of pN0 BC and 31.1% (14/45) of pN0(i+) and pN1mi BC. The presence of LVI was statistically significantly associated with worse OS, DFS, and MFS in all patients, both with and without ACt and both for ER-positive and ER-negative tumors [39]. These findings are consistent with those of other recent studies that investigated the prognostic significance of LVI [46,47,48,49,50,51,52,53,54,55,56,57]. Moreover, Gujam et al., in a systematic review concluded that LVI was a powerful prognostic factor of poorer survival, whose impact was mainly seen in patients with node-negative BC [58]. The significance of LVI as an independent prognostic factor for patients with triple-negative tumors was also reported [34,35]. In the present study, LVI was also significantly associated with a negative impact on OS, DFS, RFS, and MFS in multivariate analysis for all HER2-positive patients.

#### 4.4.1. Value of IHC Testing for ITC and Micro-Metastases in Triple-Negative and HER2-Positive BC

Detecting ITCs and micro-metastases is challenging due to their small size and limited presence in tissue samples. Conventional hematoxylin and eosin (HE) staining may lack the necessary sensitivity to accurately identify these subpopulations. Considering the prognosis significance of ITC and micro-metastases in triple-negative BC, it becomes crucial to employ more sensitive techniques, such as serial sections and IHC examination of axillary lymph nodes, to improve their detection and characterization. This is especially important for small-sized tumors without other criteria warranting ACt administration. Considering the negative impact of ITCs and micro-metastases on prognosis in our cohort, our findings suggest that the need for serial sections and IHC examination of axillary lymph nodes may also apply to HER2-positive pT1a-b breast cancer to assist in the decision-making process regarding ACt.

#### 4.4.2. Limitations

There are several limitations to our study, including its retrospective design, lack of differentiation between the types of chemotherapy used, the need for a larger cohort to analyze the impact of ER-status and LVI on pT1a-b BC, and the inability to evaluate RFS specifically for pN0(i+) and pN1mi due to the limited number of patients in this subgroup. Moreover, we were not able to report the presence of ductal carcinoma in situ (DCIS). The clinical and biological significance of HER2 overexpression in patients with DCIS remains poorly defined, and current practice guidelines do not recommend HER2 testing in DCIS patients. Nevertheless, evidence suggests that HER2-positive DCIS cases may be associated with adverse clinicopathological parameters and increased recurrence rates [59].

## 5. Conclusions

Survival outcomes were not adversely affected by pN0(i+) and pN1mi in patients with HER2-positive breast cancer. Adjuvant systemic therapy was commonly used for patients with BC ≥ pT1c. However, in the case of small HER2-positive breast cancer (pT1a-b), a negative impact on RFS was observed specifically for patients with pN0(i+) and pN1mi axillary nodes, particularly among those with pT1b tumors. These findings support the need to consider pN0(i+) and pN1mi status in the decision-making process when discussing trastuzumab-based ACt for patients with pT1b HER2-positive breast cancer.

## Figures and Tables

**Figure 1 cancers-15-04567-f001:**
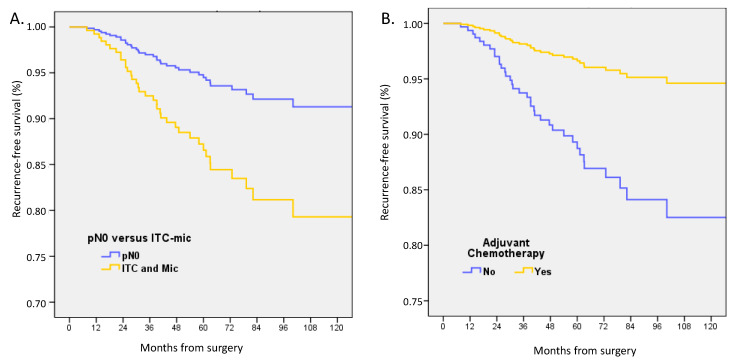
Recurrence-free survival adjusted on ER status, age, LVI, and tumor size between (**A**) patients with pT1a-b pN0 or pN0(i+)/pN1mi tumors, and (**B**) with or without adjuvant chemotherapy.

**Figure 2 cancers-15-04567-f002:**
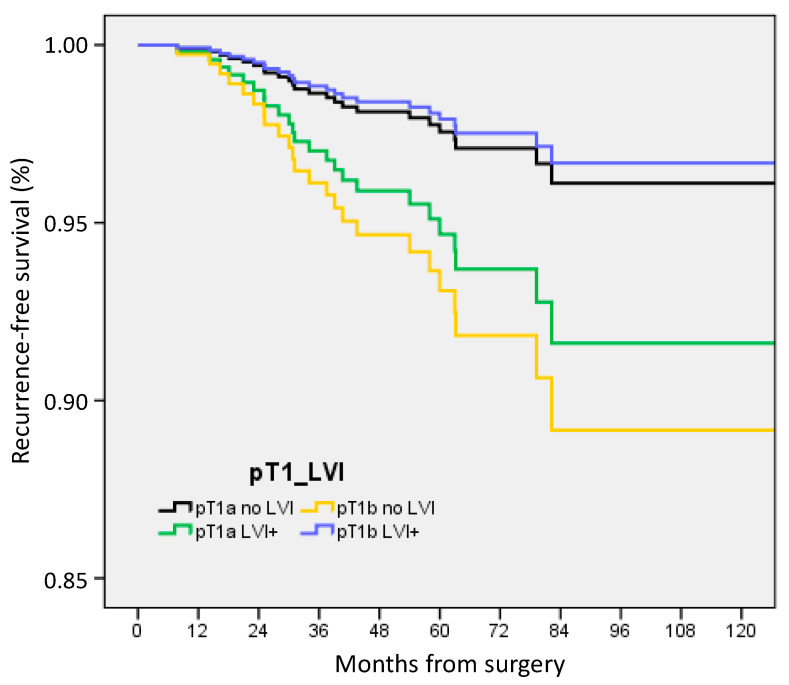
Recurrence-free survival according to tumor size and nodal status in pT1a-b patients.

**Figure 3 cancers-15-04567-f003:**
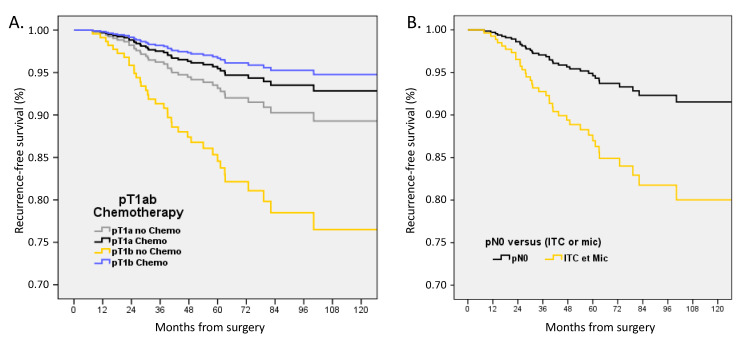
Recurrence-free survival adjusted on LVI, ER status, age groups, pN status, and pT size in (**A**) patients with pT1a-b pN0 or pN0(i+)/pN1mi tumors according to adjuvant chemotherapy, and (**B**) in patients with pT1a-b tumors according to nodal status.

**Figure 4 cancers-15-04567-f004:**
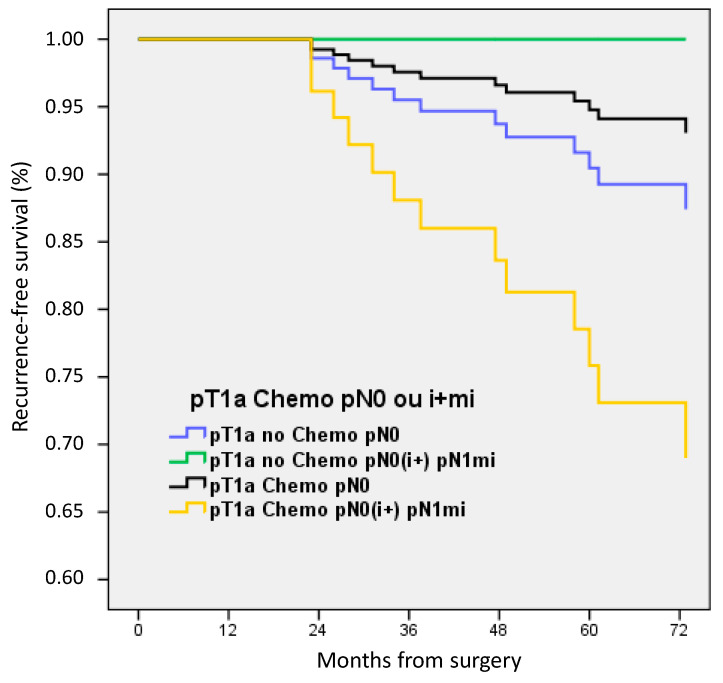
Recurrence-free survival according to adjuvant chemotherapy and nodal status in pT1a patients.

**Figure 5 cancers-15-04567-f005:**
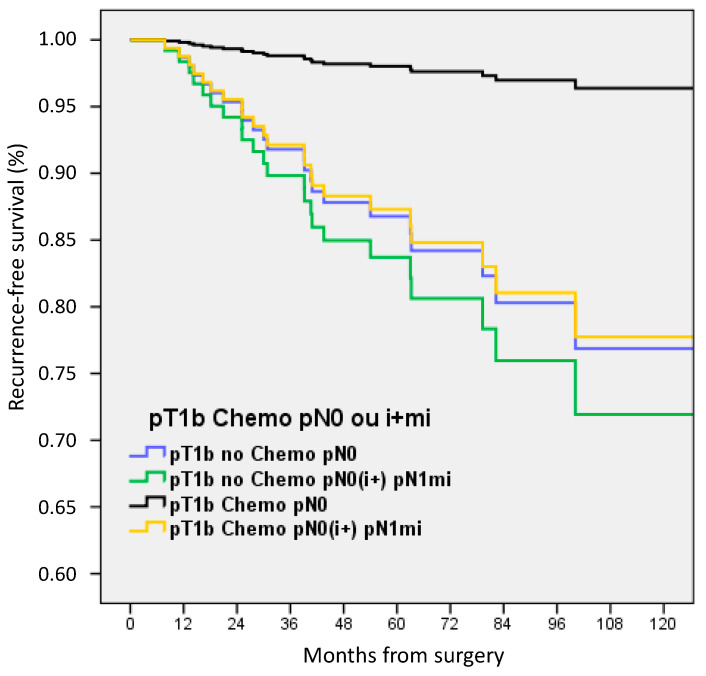
Recurrence-free survival according to adjuvant chemotherapy and nodal status in pT1b patients.

**Table 1 cancers-15-04567-t001:** Characteristics of patients according to pathologic nodal status.

All Patients Her2-Positive		pN0	pN0(i+)			pN1mi		pN1 Macro		Chi 2
		Nb	%	Nb	%	Nb	%	Nb	%	*p*
Total		1047	59.1	60	3.4	118	6.7	546	30.8	
Age	median	56		54		54.3		54		
	mean	56.2		53.4		55.2		55		
Age groups	≤40	113	10.8	12	20.0	17	14.4	77	14.1	0.007
	40.1–50	234	22.3	12	20.0	35	29.7	135	24.7	
	50.1–74.9	628	60.0	32	53.3	53	44.9	281	51.5	
	≥75	72	6.9	4	6.7	13	11.0	53	9.7	
Tumor size	median	13		15		16		23.5		
	mean	15.1		17.1		18.9		29.9		
pT size groups	<20 mm	781	74.6	35	58.3	72	61.0	194	35.5	<0.0001
	≥20 mm	240	22.9	25	41.7	43	36.4	344	63.0	
	Unknown	26	2.5	0	0	3	2.5	8	1.5	
SLNB	No	113	10.8	0	0	4	3.4	183	33.5	<0.0001
	Yes	934	89.2	60	100	114	96.6	363	66.5	
ALND	No	829	79.2	24	40.0	27	22.9	47	8.6	<0.0001
	Yes	218	20.8	36	60.0	91	77.1	499	91.4	
Grade SBR	1	103	9.8	5	8.3	11	9.3	23	4.2	<0.0001
	2	490	46.8	33	55.0	50	42.4	220	40.3	
	3	425	40.6	21	35.0	56	47.5	298	54.6	
	Unknown	29	2.8	1	1.7	1	0.8	5	0.9	
LVI	No	745	71.2	32	53.3	61	51.7	226	41.4	<0.0001
	Yes	170	16.2	25	41.7	52	44.1	283	51.8	
	Unknown	132	12.6	3	5.0	5	4.2	37	6.8	
Subtype	ER-negative	331	31.6	21	35.0	33	28.0	225	41.2	0.001
	ER-positive	715	68.4	39	65.0	85	72.0	321	58.8	
Chemotherapy	No	345	33.0	11	18.3	8	6.8	34	6.2	<0.0001
	Yes	702	67.0	49	81.7	110	93.2	512	93.8	
Trastuzumab	No	487	46.5	20	33.3	27	22.9	178	32.6	<0.0001
	Yes	560	53.5	40	66.7	91	77.1	368	67.4	
Endocrine										
Therapy	No	385	36.8	29	48.3	39	33.1	246	45.1	0.002
	Yes	662	63.2	31	51.7	79	66.9	300	54.9	
Surgery	BCS	758	72.4	35	58.3	79	66.9	260	47.6	<0.0001
	Mastectomy	268	25.6	25	41.7	39	33.1	275	50.4	
	Unknown	21	2.0	0	0	0	0	11	2.0	
Radiotherapy	No	195	18.6	11	18.3	12	10.2	23	4.2	<0.0001
	Yes (1449)	780	74.5	49	81.7	106	89.8	514	94.1	
	Unknown	72	6.9	0	0	0	0	9	1.6	
RNI *	No	477	72.5	19	50.0	26	28.9	39	8.5	<0.0001
	Yes	181	27.5	19	50.0	64	71.1	422	91.5	
Follow-up	median	58.79		62.70		67.00		63.95		
Recurrence	No	970	92.6	54	90.0	104	88.1	444	81.3	<0.0001
	Yes	77	7.4	6	10.0	14	11.9	102	18.7	
Metastases	No	1001	95.9	57	95.0	106	90.6	458	84.5	<0.0001
	Yes	43	4.1	3	5.0	11	9.4	84	15.5	
Death	No	994	94.9	58	96.7	115	97.5	470	86.1	<0.0001
	Yes	53	5.1	2	3.3	3	2.5	76	13.9	

Legend: RNI: regional nodal irradiation, * RNI 1247 known for 1449 with radiotherapy, SLNB: sentinel lymph node biopsy, ALND: axillary lymph node dissection, LVI: lympho vascular invasion, ER: endocrine receptors, BCS: breast conservative surgery.

**Table 2 cancers-15-04567-t002:** Survival results in multivariate analyses.

All Patients		Cox OS			Cox DFS			Cox RFS			Cox MFS		
		HR	CI 95%	*p*	HR	CI 95%	*p*	HR	CI 95%	*p*	HR	CI 95%	*p*
Age groups	≤40	1			1			1			1		
	40.1–50	0.842	0.41–1.74	0.643	0.767	0.49–1.21	0.256	0.721	0.45–1.15	0.172	0.852	0.49–1.47	0.566
	50.1–74.9	1.768	0.97–3.21	0.061	1.056	0.72–1.56	0.783	0.840	0.56–1.26	0.402	1.377	0.86–2.19	0.178
	≥75 ≥	4.918	2.38–10.17	<0.0001	2.157	1.30–3.58	0.003	1.265	0.71–2.26	0.427	2.796	1.57–4.99	0.001
pT size groups	<20 mm	1			1			1			1		
	≥20 mm	1.687	1.12–2.55	0.013	1.605	1.20–2.15	0.002	1.684	1.22–2.33	0.002	1.868	1.34–2.61	<0.0001
LVI	No	1			1			1			1		
	Yes	2.122	1.42–3.17	<0.0001	1.520	1.14–2.03	0.004	1.401	1.02–1.93	0.038	1.813	1.32–2.50	<0.0001
Subtype	ER-negative	1			1			1			1		
	ER-positive	0.683	0.48–0.97	0.032	0.682	0.53–0.88	0.004	0.649	0.49–0.87	0.003	0.798	0.60–1.07	0.130
pN status	pN0	1			1			1			1		
	pN0(i+)	0.440	0.11–1.84	0.261	0.780	0.34–1.80	0.560	1.120	0.48–2.61	0.793	0.494	0.15–1.59	0.236
	pN1mi	0.414	0.13–1.36	0.147	1.138	0.64–2.03	0.663	1.556	0.86–2.82	0.146	1.114	0.58–2.16	0.748
	pN1macro	1.583	1.00–2.50	0.048	1.737	1.25–2.41	0.001	2.044	1.42–2.95	<0.0001	1.824	1.26–2.64	0.001
Surgery	BCS	1			1			1			1		
	Mastectomy	1.369	0.94–2.00	0.104	1.081	0.82–1.43	0.583	1.016	0.75–1.38	0.922	1.345	0.99–1.83	0.060
	Unknown	1.806	0.64–5.06	0.261	1.126	0.49–2.58	0.780	0.980	0.36–2.69	0.970	1.633	0.70–3.78	0.252
Chemotherapy	No	1			1			1			1		
	Yes	0.536	0.33–0.87	0.012	0.500	0.36–0.70	<0.0001	0.556	0.37–0.83	0.004	0.533	0.36–0.79	0.002

Legend: LVI: lympho vascular invasion, ER: endocrine receptors, BCS: breast conservative surgery, OS: overall survival, DFS: disease free survival, RFS: recurrence free survival, MFS: metastases free survival.

**Table 3 cancers-15-04567-t003:** Characteristics of patients according to pN0 and pN0(i+) or pN1mi status.

pT1a-b		pN0		pN0(i+) & pN1mi		Chi 2
		Nb	%	Nb	%	*p*
		426	89.9	48	10.1	
Grade	1	57	13.4	5	10.4	0.887
	2	222	52.1	27	56.2	
	3	129	30.3	15	31.2	
	unknown	18	4.3	1	2.1	
Age	≤40	35	8.2	9	18.8	0.035
	40.1–50	90	21.1	14	29.2	
	50.1–74.9	279	65.5	23	47.9	
	≥75	22	5.2	2	4.2	
Subtype	ER-	169	39.7	22	45.8	0.250
	ER+	257	60.3	26	54.2	
Chemotherapy	No	218	51.2	11	22.9	<0.0001
	Yes	208	48.8	37	77.1	
Surgery	BCS	320	75.1	30	62.5	0.031
	Mastectomy	93	21.8	18	37.5	
	unknown	13	3.1	0	0	
LVI	No	308	72.3	31	64.6	<0.0001
	Yes	36	8.5	14	29.2	
	unknown	82	19.2	3	6.2	

Legend: LVI: lympho vascular invasion, ER: endocrine receptors, BCS: breast conservative surgery.

**Table 4 cancers-15-04567-t004:** Regression analyses for adjuvant chemotherapy in pT1a-b, pN0 or (pN0(i+) and pN1mi) breast cancer.

Chemotherapy: Regression		OR	CI 95%	*p*
Grade	1	1		
	2	3.834	1.83–8.04	<0.0001
	3	12.822	5.64–29.16	<0.0001
Subtype	Type_Mol(1)	2.118	1.31–3.42	0.002
Age	≤40	1		
	40.1–50	0.411	0.16–1.05	0.063
	50.1–74.9	0.273	0.12–0.65	0.003
	≥75	0.101	0.03–0.38	0.001
LVI	No	1		
	Yes	6.721	2.27–19.90	0.001
	unknown	0.334	0.18–0.60	<0.0001
pT size	pT1b	3.189	1.98–5.13	<0.0001
pN	pN0(i+) & pN1mi	2.137	0.94–4.88	0.071

**Table 5 cancers-15-04567-t005:** Multivariate survival analyses for pT1a-b pN0 or pN0(i+) and pN1mi breast cancers.

pT1a-b pN0 or pN0(i+) & pN1mi			Cox OS			Cox DFS			Cox RFS	
		HR	CI 95%	*p*	HR	CI 95%	*p*	HR	CI 95%	*p*
Age groups	≤40	1			1			1		
	40.1–50		0.00–1306	0.969	0.387	0.15–1.03	0.057	0.418	0.15–1.15	0.090
	50.1–74.9	4.134	0.42–40.7	0.224	0.445	0.20–0.98	0.043	0.336	0.14–0.80	0.014
	≥75	19.643	1.67–230.5	0.018	1.020	0.33–3.17	0.973	0.237	0.03–1.95	0.181
pT size groups	pT1a	1			1			1		
	pT1b	1.940	0.72–5.20	0.187	1.718	0.94–3.13	0.077	1.747	0.87–3.53	0.120
LVI	No	1			1			1		
	Yes	2.168	0.40–11.74	0.369	0.983	0.33–2.96	0.976	0.566	0.13–2.52	0.455
Subtype	ER-positive	1			1			1		
	ER-negative	1.246	0.50–3.09	0.635	1.677	0.95–2.97	0.077	1.667	0.84–3.31	0.144
pN status	pN0	1			1			1		
	pN0(i+)/pN1mi	0.764	0.09–6.50	0.806	1.843	0.79–4.31	0.158	2.548	1.06–6.13	0.037
Chemotherapy	No	1			1			1		
	Yes	0.259	0.07–0.97	0.046	0.272	0.13–0.56	<0.0001	0.288	0.13–0.65	0.003

## Data Availability

The datasets used and/or analyzed during the current study available from the corresponding author on reasonable request.

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
