# Peer review of "Negative Survival Impact of Occult Lymph Node Involvement in Small HER2-Positive Early Breast Cancer Treated by Up-Front Surgery"

_cancers, 2023, doi:10.3390/cancers15184567_

Round 1
Reviewer 1 Report
This manuscript introduces and comprehensively discusses the prognostic value of axillary node invasion by ITC or micro-metastases in HER2-positive BCs treated by primary surgery for all patients, as well as for patients with pT1a-b cancer.
The topic is original and relevant to the field. There is limited information on this topic in the literature.
This article makes clear the role of pN0(i+) and pN1mi status in the decision-making process when discussing trastuzumab-based ACt for patients with pT1b HER2-positive breast cancer.
There are no further improvements regarding the methodology.
-The conclusions are consistent with the evidence and arguments presented as well as summarize the main point of this article.
-References are up-to-date and appropriate
-Tables and figures are well formatted and make the study easy to follow
Minor revision
HER2 is an established prognostic and predictive marker for patients with invasive breast cancer. The clinical and biological significance of HER2 overexpression in patients with ductal carcinoma in situ (DCIS) remains poorly defined.
I would suggest a brief discussion on the clinical significance of HER2 expression in DCIS
Consider citing
https://pubmed.ncbi.nlm.nih.gov/36352293/
Author Response
Dear Editor,
Thank you for considering our work titled ““Negative survival impact of occult lymph node involvement in small HER2-positive early breast cancer treated by up-front surgery”” for potential publication in your journal.
We thank your Editorial Board and the two Reviewers for the positive and helpful comments that have been considered as described below.
Reviewer 1's Comments and Authors’ Responses:
This manuscript introduces and comprehensively discusses the prognostic value of axillary node invasion by ITC or micro-metastases in HER2-positive BCs treated by primary surgery for all patients, as well as for patients with pT1a-b cancer.
The topic is original and relevant to the field. There is limited information on this topic in the literature.
This article makes clear the role of pN0(i+) and pN1mi status in the decision-making process when discussing trastuzumab-based ACt for patients with pT1b HER2-positive breast cancer.
There are no further improvements regarding the methodology.
-The conclusions are consistent with the evidence and arguments presented as well as summarize the main point of this article.
-References are up-to-date and appropriate
-Tables and figures are well formatted and make the study easy to follow
We thank the Reviewer for his/her positive comments.
Minor revision
HER2 is an established prognostic and predictive marker for patients with invasive breast cancer. The clinical and biological significance of HER2 overexpression in patients with ductal carcinoma in situ (DCIS) remains poorly defined. I would suggest a brief discussion on the clinical significance of HER2 expression in DCIS. Consider citing https://pubmed.ncbi.nlm.nih.gov/36352293/
As suggested, a brief discussion on the clinical significance of HER2 expression in DCIS was added in the limitation section page 14 line165. The suggested citation was added.
As you can see, we have answered all the questions raised by the Reviewers and your Editorial Board and modified the manuscript as suggested. We hope this revised version will meet favorable consideration by your Editorial Board.
Sincerely,
Reviewer 2 Report
1. The information provided in the Discussion section (lines 64-92, 126-146, etc.) is difficult to perceive, it is better to present it in a tabular form.
2. The sensitivity of sentinel lymph node biopsy, according to the American Association of Clinical Oncologists (ASCO) review and clinical guidelines, ranges from 71 to 100%, the probability of obtaining a false negative result ranges from 0 to 29%, with an average of 8.4%.
A large range of values is a consequence of the use of different techniques for visualizing the sentinel lymph node, different morphological criteria for diagnosing metastasis, etc. In particular, there is still no consensus on the diagnostic value of micrometastases [0.2 - 2 mm] and single tumor cells [> 0.2 mm] in the sentinel lymph node. It would seem that the detection of tumor elements in the lymph node should be unequivocally interpreted in favor of choosing a more aggressive tactic. In the presence of single tumor cells or
micrometastases in the sentinel lymph node, the probability of metastatic damage to other lymphatic collectors is 10 and 20-35%, respectively. It should also be taken into account that the study of the sentinel lymph node is performed intraoperatively and is limited in time, and to identify such small foci, it is necessary to conduct a thorough morphological or immunohistochemical study, which cannot be implemented under conditions of intraoperative diagnosis. I also use other evaluation methods. In particular, the ratio of the level of expression of cytokeratin-19 (KRT19) mRNA and pan-leukocyte marker (CD45) in the lymph node tissue allows differential diagnosis of a metastatic and intact lymph node. There is no information on how this part is done in the peer-reviewed article. It should be added to the Materials and Methods section.
3. The results obtained by the authors cannot be associated with incorrect assignment to four groups due to the errors indicated by me above in paragraph 2?
4. N1 implies 1-3 metastases in regional lymph nodes, I did not see information on the number of affected lymph nodes in the case of all three subgroups.
Author Response
Dear Editor,
Thank you for considering our work titled ““Negative survival impact of occult lymph node involvement in small HER2-positive early breast cancer treated by up-front surgery”” for potential publication in your journal.
We thank your Editorial Board and the two Reviewers for the positive and helpful comments that have been considered as described below.
Reviewer 2's Comments and Authors’ Responses:
- The information provided in the Discussion section (lines 64-92, 126-146, etc.) is difficult to perceive, it is better to present it in a tabular form.
As suggested, two supplementary tables (3 & 4) were added in the discussion section page 12 line 69 and page 14 line 134
- The sensitivity of sentinel lymph node biopsy, according to the American Association of Clinical Oncologists (ASCO) review and clinical guidelines, ranges from 71 to 100%, the probability of obtaining a false negative result ranges from 0 to 29%, with an average of 8.4%.
A large range of values is a consequence of the use of different techniques for visualizing the sentinel lymph node, different morphological criteria for diagnosing metastasis, etc. In particular, there is still no consensus on the diagnostic value of micrometastases [0.2 - 2 mm] and single tumor cells [> 0.2 mm] in the sentinel lymph node. It would seem that the detection of tumor elements in the lymph node should be unequivocally interpreted in favor of choosing a more aggressive tactic. In the presence of single tumor cells or micrometastases in the sentinel lymph node, the probability of metastatic damage to other lymphatic collectors is 10 and 20-35%, respectively. It should also be taken into account that the study of the sentinel lymph node is performed intraoperatively and is limited in time, and to identify such small foci, it is necessary to conduct a thorough morphological or immunohistochemical study, which cannot be implemented under conditions of intraoperative diagnosis. I also use other evaluation methods. In particular, the ratio of the level of expression of cytokeratin-19 (KRT19) mRNA and pan-leukocyte marker (CD45) in the lymph node tissue allows differential diagnosis of a metastatic and intact lymph node. There is no information on how this part is done in the peer-reviewed article. It should be added to the Materials and Methods section.
All sentinel lymph node biopsies were analyzed by serial sections with standard HES after fixation. No intraoperatively analysis were performed. If all the serial sections were negative, an additional IHC analysis was carried out. No other technics such One-step nucleic acid amplification (OSNA) was performed. The following sentence was added in the methods section to improve the understanding of pathological assessment page 3 line 99 :” All sentinel lymph node biopsies were analyzed by serial sections with standard HES after fixation. No intraoperatively analysis were performed. If all the serial sections were negative, an additional IHC analysis was carried out”.
- The results obtained by the authors cannot be associated with incorrect assignment to four groups due to the errors indicated by me above in paragraph 2?
As mentioned by the reviewer, intraoperatively SN analysis is limited. In our study, all sentinel lymph node biopsies were analyzed by serial sections with standard HES after fixation. No intraoperatively analysis were performed. If all the serial sections were negative, an additional IHC analysis was carried out
- N1 implies 1-3 metastases in regional lymph nodes, I did not see information on the number of affected lymph nodes in the case of all three subgroups.
Our study focus on the prognostic value of axillary node invasion by ITC or micro-metastases. The N1macro group implies indeed 1-3 metastases in regional lymph nodes, but represent only the control group of the main cohort.
As you can see, we have answered all the questions raised by the Reviewers and your Editorial Board and modified the manuscript as suggested. We hope this revised version will meet favorable consideration by your Editorial Board.
Sincerely,
Round 2
Reviewer 2 Report
I have no more comments on the article; in its present form, the manuscript can be recommended for publication.